# Mitogenomic Characterization and Comparative Analysis of Three Egg Parasitoid Wasps Parasitizing *Nilaparvata lugens* (Stål)

**DOI:** 10.3390/insects16050543

**Published:** 2025-05-20

**Authors:** Wei He, Tingting Li, Liyang Wang, Hongxuan Wu, Jie Wang, Qiang Zhou

**Affiliations:** State Key Laboratory of Biocontrol, School of Life Sciences, Sun Yat-sen University, Guangzhou 510275, China; hewei83@mail2.sysu.edu.cn (W.H.); 13553837997@163.com (T.L.); wangly@stu.ynu.edu.cn (L.W.); wuhx69@mail2.sysu.edu.cn (H.W.); wangj979@mail2.sysu.edu.cn (J.W.)

**Keywords:** mitochondrial genome, phylogenetic analysis, *Anagrus*, *Pseudoligosita*

## Abstract

Egg parasitoid wasps play a key role in the biological control of *Nilaparvata lugens* (Stål), a major rice pest in Asia. In this study, we sequenced and analyzed the complete mitochondrial genomes of three such parasitoids—*Pseudoligosita nephotetticum*, *Anagrus frequens*, and *Anagrus nilaparvatae*—for the first time. The genomes exhibited high A + T content and diverse gene arrangements, with *P. nephotetticum* showing extensive rearrangements compared to the more conserved *Anagrus* species. The non-synonymous (Ka) to synonymous (Ks) substitution ratio analysis revealed signals of adaptive evolution, particularly in genes related to energy metabolism. Phylogenetic trees constructed from 46 Chalcidoidea mitogenomes revealed the relationship of the genus *Anagrus* within Mymaridae and supported the evolutionary distinctiveness of *Pseudoligosita* within Trichogrammatidae.

## 1. Introduction

Brown rice planthopper (*Nilaparvata lugens* Stål) is a major pest of rice crops across many Asian countries [1]. Current control strategies rely predominantly on traditional chemical insecticides, although the deployment of egg parasitoid wasps as biological control agents has also yielded promising results [2]. A total of 56 egg parasitoid species capable of parasitizing *N. lugens* eggs have been recorded in the Asian rice ecosystem, belonging to 17 genera within four families [3]. Among these, members of the families Mymaridae and Trichogrammatidae have received the most research attention. Both families include minute, metallic-sheen-lacking egg parasitoids that exhibit considerable morphological and biological similarities [4]. Mymaridae primarily parasitize hosts from Hemiptera, such as *Dalbulus maidis*, *Peregrinus maidis*, and *N. lugens* [5], whereas Trichogrammatidae tend to target Lepidopteran species, including *Helicoverpa armigera*, *Ostrinia furnacalis*, and *Argyroploce schistaceana* [6]. In southern China, only five species from the genus *Anagrus* (family Mymaridae) have been clearly identified as parasitoids of Hemipteran rice pests, with their effectiveness validated in laboratory or field conditions: *A*. *nilaparvatae* [7], *A*. *frequens* [8], *A*. *perforator* [9], *A*. *optabilis* [8], and *A*. *flaveolus* [10]. In addition, two species from the genus *Pseudoligosita* (family Trichogrammatidae) have also been confirmed: *P*. *yasumatsui* [11] and *P*. *nephotetticum* [12].

The Chalcidoidea is the most taxonomically diverse superfamily within Hymenoptera, with over 22,000 species currently described and estimates suggesting that the total number may exceed 500,000 [4]. Most of them are parasitoids, and the minute body size of many species renders this superfamily one of the most taxonomically challenging groups. The phylogenetic classification of Chalcidoidea has significantly progressed in recent years. Earlier efforts primarily focused on morphological traits and single-gene analyses, but with the advent of high-throughput sequencing, transcriptomic data are increasingly used to explore Chalcidoid phylogenetics [13,14]. Among available molecular markers, the mitochondrial genome is frequently utilized due to its conserved gene order, maternal inheritance, compact structure, and rapid evolutionary rate [15,16]. As of January 2025, nine Trichogrammatinae with fully annotated mitochondrial genomes have been uploaded to NCBI, seven of which are from the genus *Trichogramma* [17,18,19], with the remaining two belonging to *Megaphragma* [20] and *Pseudoligosita* [21], respectively. In contrast, the only complete mitochondrial genome of Mymaridae currently available on NCBI is that of *Gonatocerus* sp. [22]. In this study, we sequenced the complete mitochondrial genomes of *P. nephotetticum*, *A. frequens*, and *A. nilaparvatae* using both Sanger sequencing and high-throughput sequencing technologies. We analyzed nucleotide composition, codon usage, and tRNA secondary structures to compare mitochondrial genome characteristics among these species. This study also aims to determine whether *Pseudoligosita*, a genus of Trichogrammatidae capable of parasitizing *N. lugens*, shares molecular similarities with *Anagrus*. Furthermore, to elucidate the phylogenetic placement of these taxa, we reconstructed phylogenetic trees based on PCG123 (all three codon positions) and PCG12 (first and second codon positions only) datasets using Maximum Likelihood (ML) and Bayesian Inference (BI) approaches. A total of 46 annotated mitogenomes from Chalcidoidea were included as ingroups, while two mitogenomes from Cynipoidea served as outgroups, providing robust data for resolving the phylogenetic position of these families and their internal relationships. After 2–3-years, a laboratory-based field survey of parasitoid population dynamics revealed *P. nephotetticum*, *A. frequens*, and *A. nilaparvatae* as the predominant egg parasitoids of *N. lugens* in rice agroecosystems. Among them, *A. nilaparvatae* is considered a dominant species due to its high fecundity and short developmental period [11]. A comprehensive mitogenomic analysis of these field-collected parasitoids can enrich the phylogenetic framework of Chalcidoidea and contribute to our understanding of their evolutionary relationships. Moreover, such studies provide molecular tools for investigating host–parasitoid coevolution, enabling rapid species identification and population monitoring in the field, and facilitating the selection of highly effective parasitoid strains for biological control of *N. lugens*.

## 2. Materials and Methods

### 2.1. Sample Collection and DNA Extraction

*P. nephotetticum*, *A. frequens*, and *A. nilaparvatae* were all collected from rice fields at South China Agricultural University (SCAU), Guangdong Province, China (23.1622339° N, 113.360629° E), using *N. lugens* egg to entrap. The body length of *P. nephotetticum* ranges from 0.60 to 0.66 mm. The general coloration is yellowish-brown, while the compound eyes and ocelli are black. The vertex, venation of forewings and hindwings, and posterior halves of tergites on abdominal segments 1–4 exhibit a pale yellowish-brown hue. Segments posterior to the fifth abdominal segment are distinctly brown. The entire surface of the forewings appears hyaline and devoid of pigmentation [12]. The body color of *A. frequens* is yellow-brown. The antennae are geniculate, composed of a scape, pedicel, and flagellum, with the flagellum terminating in an enlarged clava. A single line of cilia is present along both the anterior and posterior margins of the forewing blade. The upper surface of the forewing lacks setae. The general body color of *A. nilaparvatae* is yellow-brown, with the scape, pedicel, and postscutellum distinctly yellow. The forewing is broadened in the distal third and bears a mid-longitudinal row of discal cilia extending from the distal portion of the stigmal vein to the wing apex. Additional scattered cilia are present between this row and both the anterior and posterior margins, resulting in the absence of any hairless area [23]. The next generation was reared in the laboratory and identified based on morphological characteristics [12,23]. Before DNA extraction, specimens were preserved in 90–95% ethanol and stored at −20 °C. Genomic DNA was extracted non-destructively from whole adult wasps following the protocol of Taekul et al. (2014) [24]. Extracted DNA was stored at −20 °C according to the manual of the DNeasy Blood & Tissue Kit (QIAGEN GmbH, Hilden, Germany).

### 2.2. Sanger Sequencing

Referring to the *cox1* universal primer [25] and universal animal mt primers [15], primers (Appendix A) were designed based on gene units to obtain the whole mitochondrial DNA of *P. nephotetticum* and *A. frequens*. PCR was performed with Tks Gflex DNA Polymerase (Takara Bio Inc., Kusatsu, Japan). In this study, each sample consisted of a single adult wasp, and all individuals within a species were offspring of the same female. A total of 15 individuals per species were used for sequencing. The 50 μL PCR reaction mixture contained 25 μL of 2 × Gflex PCR Buffer (Mg^2+^, dNTP plus), 2 μL F primer (10 μM), 2 μL R primer (10 μM), 5 μL of DNA and 16 μL of water. The PCR cycling conditions were as follows: activation for 1 min at 98 °C, followed by 35 cycles of 15 s at 98 °C, 50 s at 50–52 °C, and 1 min at 72 °C. The final cycle was followed by an extension of 5 min at 72 °C. PCR products were analyzed by electrophoresis on a 1.5% (g/mL) agarose gel for 20 min at 140 V. All PCR products were purified using the Cycle-Pure Kit (100) (Omega Bio-Tek, Norcross, GA, USA) and sent to The Beijing Genomics Institute (BGI) (Shenzhen, China) for Sanger sequencing.

### 2.3. High-Throughout Sequencing

Due to the high AT content in certain regions, complete mitogenome sequences for *A. frequens* could not be obtained via Sanger sequencing. Consequently, next-generation sequencing (NGS) was employed for *A. frequens* and *A. nilaparvatae*. Genomic DNA was fragmented into 300–500 bp fragments using a Covaris S220 Focused Ultrasonicator (Covaris, Woburn, MA, USA). Library preparation was performed using the TruSeq™ Nano DNA Sample Prep Kit (Illumina, Inc., San Diego, CA, USA), which included the blunt ends repair, A-tailing, adapter ligation, purification, and PCR amplification. Sequencing was conducted on the Illumina HiSeq X platform. Raw sequencing reads were processed using Trimmomatic-0.39 to remove adapters and low-quality sequences, generating clean data (BIOZERON Co., Ltd., Shanghai, China, Tianjin Biochip Corporation, Tianjin, China).

### 2.4. Mt Genome Sequencing, Annotation and Bioinformatic Analyses

Short reads obtained from Sanger sequencing and NGS were assembled into complete circular mitochondrial genomes using Geneious v9.0.2 [26] and SPAdes v3.10.1 [27]. For the two *Anagrus* species, analyses were conducted using *Gonatocerus* sp. (accession number: MF776883), the only member of Mymaridae with a published complete mitochondrial genome, as the primary reference. In the case of *P*. *nephotetticum*, *P*. *yasumatsui* (accession number: NC_082212) was used as the main reference sequence, supplemented with additional species from Trichogrammatidae for comparative analysis. Mitogenomes were annotated using the MITOS Web Server (http://mitos.bioinf.uni-leipzig.de/index.py, accessed on 5 November 2024) to identify protein-coding genes (PCGs), ribosomal RNA (rRNA) genes, transfer RNA (tRNA) genes, and their secondary structures. Unannotated tRNA genes were identified using tRNA-scan (https://github.com/UCSC-LoweLab/tRNAscan-SE, accessed on 19 November 2024) and RNA structure (https://rna.urmc.rochester.edu/RNAstructure.html, accessed on 19 November 2024). Gene boundaries for 13 PCGs and two rRNAs were manually curated through sequence alignment with publicly available mitogenomes. Genomic characteristics, including total sequence length, nucleotide composition, codon usage, and amino acid distribution, were analyzed using Geneious v9.0.2 and MEGA v11.0.13 [28]. R v4.4.0 [29] was used to generate relative synonymous codon usage (RSCU) plots for each genus. DnaSP v6.0 [30] was used to calculate the rates of non-synonymous substitution rate (Ka)/synonymous substitution rate (Ks) for each PCG. AT and GC skews were calculated using the formulas: AT skew = (A − T)/(A + T) and GC skew = (G − C)/(G + C) [31]. Mitochondrial genome maps were visualized using Proksee (https://proksee.ca/, accessed on 2 January 2025).

### 2.5. Gene Arrangement and Phylogenetic Analyses

The three mitochondrial genomes obtained in this study were compared with the putative ancestral gene order of insects, represented by *Drosophila yakuba* (accession number: NC_001322) [32]. Additionally, *Gonatocerus* sp. and eight parasitoid species from Trichogrammatidae were included for comparative analysis.

A total of 13 PCGs from 48 mitochondrial genomes were selected for phylogenetic analysis, including 46 Chalcidoidea species and two outgroup species from Cynipoidea: *Trichagalma acutissimae* and *Ibalia leucospoides* (Appendix A) [33,34,35]. Protein sequences were aligned using Gblocks v0.91b [36] to remove poorly aligned positions and gaps, generating a refined SuperMatrix dataset. The optimal evolutionary models for individual genes were determined using MEGA v11.0.13 (Appendix A), and concatenated alignments were performed using BioEdit v7.0.9.0 [37]. Two datasets were generated for phylogenetic reconstruction: PCG123 (all three codon positions) and PCG12 (first and second codon positions only). Substitution saturation tests were performed on both the PCGs and PCG12 datasets using DAMBE v7.0.35 [38]. Sequence heterogeneity of both the PCGs and PCG12 datasets was assessed using the ALiGROOVE [39]. Phylogenetic relationships were inferred using Maximum Likelihood (ML) and Bayesian Inference (BI). BI analyses were conducted using MrBayes v3.2.7 [40], with 10 million generations, sampling every 1000 generations, and a 25% burn-in. ML analyses were performed using IQ-TREE [41], which automatically identified the optimal substitution model and applied the ultrafast bootstrap (UFB) algorithm with 10,000 replicates to assess nodal support. The resulting phylogenetic trees were visualized and refined using FigTree v1.4.0 [42].

## 3. Results

### 3.1. Genome Structure, Organization and Composition

The complete mitochondrial genomes of *P. nephotetticum*, *A. frequens*, and *A. nilaparvatae* were 15,889 bp, 15,510 bp, and 15,429 bp in length, respectively (Appendix A). The mitogenome of *P. nephotetticum* contained 37 genes (13 PCGs, 22 tRNAs, and two rRNAs) along with an AT-rich region. Of these, 27 genes were encoded on the heavy strand and 10 on the light strand. In contrast, *A. frequens* and *A. nilaparvatae* each contained only 36 genes, with the *trnL1* gene not successfully identified in either species. In both *Anagrus* species, the AT-rich region was located between *trnG* and *trnV*, while in *P. nephotetticum*, it was positioned between *rrnS* and *trnV*, with a length of 395 bp. Circular mitochondrial genome maps marking all genes and AT-rich regions are presented in Figure 1A–C. The complete mitochondrial genomes of *P. nephotetticum*, *A. frequens*, and *A. nilaparvatae* exhibited a strong A + T bias, with overall A + T contents of 86.6%, 86.9% and 86.9%, respectively (Table 1). The protein-coding genes (PCGs) consistently showed the lowest A + T content compared to tRNA and rRNA regions (84.9%/85.2%/84.6%). Among the PCGs, the second codon positions displayed the lowest A + T content (81.7%/82.0%/80.7%), whereas the third codon positions had the highest (87.8%/88.9%/87.2%). All three complete mitogenomes exhibited negative AT skew values (−0.11/−0.03/−0.06) and positive GC skew values (0.20/0.06/0.05), suggesting a compositional bias toward T over A and G over C.

### 3.2. Protein-Coding Genes

The PCGs spanned 11,113 bp (*P. nephotetticum*), 11,135 bp (*A. frequens*), and 11,129 bp (*A. nilaparvatae*). The longest gene in all three species was *nad5* (ranging from 1669 to 1717 bp), while the shortest was *atp8* (154–162 bp) (Appendix A). The predicted initiation codons followed the ATN pattern, with three types identified: ATG, ATT, and ATA. The most frequently used start codons were ATG and ATT in *P. nephotetticum*, while ATA was predominant in both *Anagrus* species. In *P. nephotetticum*, all PCGs terminated with the standard stop codon TAA. In contrast, *A. nilaparvatae* utilized TAG as the stop codon for *atp6*, and the incomplete stop codon T for both *atp8* and *nad6*. Similarly, *A. frequens* exhibited two alternative termination codons in addition to TAA, with TAG terminating *nad6* and T serving as the stop codon for *atp8* and *nad5*.

The mitochondrial PCGs of *P. nephotetticum*, *A. frequens*, and *A. nilaparvatae* comprised 3719, 3424, and 3486 codons, respectively. The analysis of relative synonymous codon usage (RSCU) revealed that the three most frequently used codons in *P. nephotetticum* were UUU (Phe), UUA (Leu2), and AUU (Ile) (Figure 2A), with usage counts of 424, 402, and 403, respectively (Appendix A). In contrast, the most frequently used codons in *A. frequens* and *A. nilaparvatae* were AUU (Ile), UCU (Ser2), and UUA (Leu2) (Figure 2B,C). All of these frequently used codons end with either A or U at the third codon position, indicating a codon usage bias toward A or U in the third base. This pattern also reflects the high abundance of T and A nucleotides throughout the mitochond.

The evolutionary rates of 13 PCGs in three Mymaridae species and nine Trichogrammatidae species were assessed using the non-synonymous (Ka) to synonymous (Ks) substitution ratio analysis, with *D. yakuba* as a reference. In *Anagrus* species, seven genes exhibited Ka/Ks values greater than 1, suggesting positive selection, with *atp8* displaying the highest evolutionary rate (Ka/Ks = 3.14) (Figure 3A). Similarly, in *Pseudoligosita* and *Trichogramma* species, six genes were under positive selection, with *nad6* showing the highest evolutionary rate (Ka/Ks = 3.40 and 2.93, respectively) (Figure 3B,C). In contrast, *cox1* exhibited the lowest evolutionary rate across all three genera (Ka/Ks = 0.36–0.42), indicating strong purifying selection (Appendix A).

### 3.3. The tRNAs and rRNAs

The *P. nephotetticum* mitogenome contained 22 tRNA genes, with 14 encoded on the heavy strand and 8 on the light strand, ranging in length from 63 to 71 bp. All tRNAs exhibited the typical cloverleaf structure except for *trnS1*, which lacked the DHU arm. A total of 10 G-U mismatches were identified, distributed across the amino acid acceptor arm (six), DHU arm (three), and anticodon arm (one) (Appendix A). In contrast, *A. frequens* and *A. nilaparvatae* contained only 21 tRNA genes, accounting for 9.25% and 9.1% of their mitochondrial genomes, respectively. A total of 19 tRNAs exhibited the typical cloverleaf structure, while *trnS1* and *trnR* lacked the DHU arm. The *trnL1* gene could not be identified based on secondary structure prediction in either of the *Anagrus* species. Secondary structure analyses revealed 12 and 9 mismatched base pairs in *A. frequens* and *A. nilaparvatae*, respectively, all being G-U mismatches (Appendix A). The mitogenomes contained two rRNAs, with lengths of 2131 bp (*P. nephotetticum*), 1964 bp (*A. frequens*), and 2053 bp (*A. nilaparvatae*), exhibiting AT contents of 88.6–91.7% (Table 1). In *P. nephotetticum*, *rrnS* was located between *trnV* and *nad3*, and *rrnL* between *trnA* and *trnL1*. In *A. frequens* and *A. nilaparvatae*, *rrnL* and *rrnS* were positioned between *nad1-trnA* and *trnA*-(AT-rich region), respectively. A notable difference between the two species lies in the AT skew of the complete rRNA genes: *A. frequens* exhibited a negative AT skew (−0.008), whereas *A. nilaparvatae* showed a positive value (0.090). This indicates that in the rRNA sequences of *A. frequens*, thymine (T) is used more frequently than adenine (A), whereas A is used more frequently than T in *A. nilaparvatae*.

### 3.4. Gene Rearrangement

To investigate gene rearrangement in three parasitoid wasps, we referenced the mitochondrial genome of *D. yakuba* (Figure 4). *P. nephotetticum* exhibited extensive rearrangements in both protein-coding genes (PCGs) and transfer RNA (tRNA) genes. The primary characteristic of PCGs rearrangement is inversion, notably involving the transposition of *nad2* and the large-scale segmental inversion of *nad5-nad4-nad4L-nad6-cob-nad1*, although the relative positions of these genes remain unchanged. tRNA genes exhibit the most serious rearrangements. These rearrangements are primarily concentrated within the *trnI-trnO-trnM-trnW-trnC-trnY* gene cluster. However, additional tRNAs rearrangements also occur outside this region. For instance, within the conserved gene block (*cox1-trnL2-cox2-trnK-trnD-atp8-atp6-cox3*), *trnK* underwent inversion and relocates downstream of *trnD*. In contrast, *A. frequens* and *A. nilaparvatae* exhibited a relatively conserved gene arrangement, with 13 PCGs maintaining the ancestral order. However, tRNA genes underwent moderate rearrangement. Specifically, *trnQ* and *trnM* were inverted and repositioned between *trnF* and *nad5*, *trnE*, *trnV*, and *trnI* were inverted and relocated, and *trnA* was translocated, with *trnL1* absent. The major rearrangements were concentrated within the *trnE-trnF-trnH* and *trnV-trnI-trnQ-trnM* gene clusters.

### 3.5. Phylogenetic Analyses

In this study, phylogenetic relationships within the superfamily Chalcidoidea were reconstructed based on three newly sequenced mitochondrial genomes of egg parasitoid wasps, along with 43 previously published mitogenomes from GenBank. Two species from Cynipoidea, *T. acutissimae* and *I. leucospoides*, were selected as outgroups. Compared to the first and second codon positions, the third position is typically more variable because of codon degeneracy, leading to a higher rate of synonymous substitutions and a greater likelihood of substitution saturation. To evaluate the suitability of our datasets for phylogenetic reconstruction, we performed substitution saturation tests on both PCGs and PCG12 datasets. The results showed that, for the PCGs dataset, Iss = 0.360 < Iss.c = 0.855 with *p* < 0.05, and for PCG12, Iss = 0.340 < Iss.c = 0.851 with *p* < 0.05, indicating that both datasets are appropriate for phylogenetic analysis (Iss = Index of Substitution Saturation, Iss.c = Critical Value of the Index of Substitution Saturation). The heterogeneity test results indicated that both datasets exhibited a certain degree of heterogeneity, with overall differences being relatively small. However, in a few species, PCG12 showed lower heterogeneity than PCGs. This suggests that the higher A + T content at the third codon position may have contributed to the increased sequence heterogeneity (Appendix A). Phylogenetic analyses based on the concatenated PCG123 and PCG12 datasets were performed using both Bayesian inference (BI) and maximum likelihood (ML) methods (Figure 5A,B; Appendix A). In the phylogenetic analyses of Chalcidoidea based on 13 mitochondrial protein-coding genes, all resulting trees (both ML and BI) consistently supported Mymaridae as occupying the basal position within the superfamily, suggesting that it represents the earliest-diverging monophyletic lineage of Chalcidoidea. In contrast, Chalcididae was recovered at the terminal position across all tree topologies, indicating its status as a derived group within the superfamily. Topological inconsistencies were observed among the remaining Chalcidoid families, depending on dataset (PCG123 vs. PCG12) and analytical method (ML vs. BI). For example, the sister-group relationship between Aphelinidae and Trichogrammatidae was stable in the PCG123 dataset but was not retained in PCG12. In the PCG123 dataset, both ML and BI analyses supported the basal clade formed by Eurytomidae + Torymidae, which was subsequently joined by Encyrtidae to form a monophyletic group. However, this clade was merged into a broader backbone clade in the BI analysis of PCG12. Additionally, Agaonidae was positioned upstream of the Pteromalidae + Eulophidae clade in the PCG123 dataset, forming a paraphyletic group, while inconsistent relationships were observed in the PCG12 trees. Despite these variations, the internal relationships within Mymaridae and Trichogrammatidae were congruent across all four phylogenetic trees. The results consistently supported the following tree topology: (*A. frequens* + *A. nilaparvatae*) + *Gonatocerus* sp.; ((*P. yasumatsui + P. nephotetticum*) + *M. amalphitanum*) + the remaining Trichogrammatidae taxa included in our dataset.

## 4. Discussion

The complete mitogenomes of *P. nephotetticum*, *A. frequens*, and *A. nilaparvatae* measured 15,889 bp, 15,510 bp, and 15,429 bp, respectively, consistent with the typical insect mitochondrial genome size range of 14–20 kb [43]. The AT content of the three mitogenomes was relatively high (86.6%, 86.9%, and 86.9%, respectively), a characteristic commonly observed in Hymenoptera mitochondrial genomes [44,45]. Although insect mitogenomes typically exhibit positive AT skew and negative GC skew [46], our results revealed a negative AT skew and a positive GC skew, suggesting a preference for thymine (T) and guanine (G) bases in these parasitoid wasps. Similar AT and GC skew bias have been reported in other Hymenopteran species [31].

Most tRNA genes in the three species followed the expected cloverleaf structure, except for *trnS1* and *trnR*, which lacked the DHU arm (Appendix A). The absence of the DHU arm in *trnS1* is a well-documented phenomenon in wasps [17,22,34]. However, in *A. frequens* and *A. nilaparvatae*, *trnR* also lacked the DHU arm, and *trnL1* could not be annotated, a condition previously reported only in *Gonatocerus* sp. [22]. The failure to annotate *trnL1* was most likely due to its low primary sequence conservation or the presence of atypical secondary structures [47]. The combination of undetectable *trnL1* and the atypical *trnR* structure observed in both *Anagrus* species may represent a derived condition within Mymaridae, and merits further investigation. In contrast to the typical light-strand location of rRNAs in Chalcidoidea [33], both rRNAs in *P. nephotetticum* were located on the heavy strand, consistent with prior findings for *P. yasumatsui* [21].

In the analysis of PCGs, both *Anagrus* species were found to contain genes ending with the incomplete stop codon “T”, a feature also reported in the mitochondrial genome of the congeneric *Gonatocerus* sp., and commonly observed in invertebrate mitochondrial genomes [48,49]. The non-synonymous (Ka) to synonymous (Ks) substitution ratio analysis revealed that *atp8*, *nad2*, *nad4l*, and *nad6* were under positive selection, while *nad4* and *nad5* exhibited Ka/Ks values near 1, indicating neutral selection. Notably, *nad3* and *nad4* were subject to positive selection in *Anagrus* and *Pseudoligosita* but experienced either neutral or purifying selection in Trichogramma. This could be attributed to differences in oviposition behavior, as *N. lugens* eggs are smaller and embedded individually within rice sheaths or stems, requiring greater energy expenditure for host searching [50]. Both genes encode components of the mitochondrial NADH dehydrogenase complex (Complex I), which plays a critical role in proton pumping and ATP production [51]. Positive selection on these genes suggests adaptive evolution, possibly in response to ecological pressures, leading *Pseudoligosita* and *Anagrus* to exhibit similar evolutionary trends.

Gene rearrangement phenomena are common in Hymenopteran mitochondrial genomes [52,53], primarily involving tRNA genes but also occurring in PCGs and rRNA genes, with inversions being the predominant rearrangement type. Mitochondrial gene rearrangement is considered a valuable phylogenetic trait in metazoans [52]. The patterns observed in this study, including species from the genera *Anagrus*, *Pseudoligosita*, and *Trichogramma*, indicate that gene order is generally conserved within genera [54]. The variation in mitochondrial gene arrangements among different genera offers potential for the design of genus-specific or species-specific primers at unique rearrangement sites for multiplex PCR-based identification [55]. All three genera exhibited gene arrangements distinct from the ancestral insect mitochondrial genome, with *Pseudoligosita* and *Trichogramma* (When the orientation of *cox1* is set as the forward strand, seven parasitoid species from the genus *Trichogramma* share a consistent gene arrangement pattern, hereafter referred to as the “*Trichogramma*” arrangement) showing uncommon rearrangement patterns rarely reported in Chalcidoidea [56]. The extensive gene rearrangements observed in *Pseudoligosita*, in contrast to the conserved architecture of *Anagrus*, suggest that mitochondrial genome organization may provide a useful tool for resolving cryptic species complexes within Trichogrammatidae. *A. frequens* and *A. nilaparvatae* displayed identical gene rearrangement patterns, differing from *Gonatocerus* sp. Further mitogenomic data are needed to determine whether these rearrangements are unique to *Anagrus*.

Phylogenetic analyses based on 13 PCGs strongly supported the monophyly of Trichogrammatidae [18,19] and Mymaridae, with Mymaridae consistently positioned at the basal lineage of Chalcidoidea. However, the sister-group relationship of Trichogrammatidae remains uncertain across studies. In our ML and BI phylogenies, Aphelinidae and Trichogrammatidae were recovered as sister groups [33,35], whereas alternative datasets have placed Trichogrammatidae as the sister group to Pteromalidae [33].

The conflicting sister-group relationships may reflect differences in taxon sampling or long-branch attraction artifacts. In recent years, with the advancement and widespread application of second-generation and third-generation sequencing technologies, increasingly large datasets have been employed to explore phylogenetic relationships within Chalcidoidea. Transcriptomic data have become a particularly valuable resource for reconstructing internal relationships in this highly diverse superfamily. Early phylogenomic studies analyzed 3239 single-copy orthologs from 37 chalcidoid taxa and 11 outgroups, but support for deep relationships along the chalcidoid backbone remained weak, with only the grouping of Mymaridae + (Trichogrammatidae + other Chalcidoidea) being consistently recovered [13]. Subsequent studies incorporated broader taxon sampling and more extensive transcriptomic data, yielding a more resolved topology: (Mymaridae + (((Trichogrammatidae + Eulophidae) + (Encyrtidae + Aphelinidae)) + remaining Chalcidoidea)) [14]. In addition, other approaches have been attempted for phylogenetic relationships within the Chalcidoidea, such as the construction of phylogenetic relationships in the Chalcidoidea based on ultraconserved elements (UCEs) and exons [57]. Despite these advances, many relationships within Chalcidoidea remain unresolved and sometimes conflicting, highlighting the need for continued data collection and methodological refinement to achieve a widely accepted phylogenetic framework. Future phylogenomic analyses with broader taxon representation are needed to resolve this discrepancy.

This study represents the first complete mitochondrial genome sequencing of *Anagrus* species. Additional mitogenomic data will be necessary to refine the phylogenetic placement of *Anagrus* within Mymaridae. Currently, only one mitochondrial genome has been reported for *Pseudoligosita*, that of *P. yasumatsui*. The observed phylogenetic relationships in this study ((*Pseudoligosita* + *Megaphragma*) + *Trichogramma*) agree with previous findings [21], further supporting the evolutionary distinctiveness of *Pseudoligosita* within Trichogrammatidae.

## 5. Conclusions

This study presents the first complete mitochondrial genomes of three egg parasitoid wasps targeting *N. lugens*—*P. nephotetticum*, *A. frequens*, and *A. nilaparvatae*—and provides novel insights into their genomic architecture and phylogenetic placement within Chalcidoidea. All three mitogenomes exhibit high A + T content, typical gene compositions, and strand-specific nucleotide skews, with notable variations in codon usage and evolutionary rates. Extensive gene rearrangements were detected in *P. nephotetticum*, while both *Anagrus* species displayed conserved gene orders, supporting genus-level stability in mitochondrial architecture. Structural differences in tRNA genes, such as the failure to annotate *trnL1* and the presence of an atypical *trnR*, may reflect derived characteristics within Mymaridae and merit further study. Phylogenetic reconstructions based on 13 PCGs strongly supported the grouping of (*Pseudoligosita* + *Megaphragma*) + *Trichogramma* aligns with previous molecular evidence and underscores the distinct evolutionary trajectory of *Pseudoligosita*. This study’s identification of genus-specific gene rearrangements and positively selected genes not only reveals potential molecular markers for designing parasitoid strains with enhanced host-selection efficiency, but also explores molecular tools to investigate host-parasitoid coevolutionary dynamics, thereby facilitating rapid field identification of species, continuous population tracking, and precision selection of optimized biocontrol agents against *N. lugens*. In addition, these findings expand the mitochondrial genome resources available for Chalcidoidea, refine current phylogenetic frameworks, and offer a valuable foundation for future research on parasitoid evolution, taxonomy, and biological control strategies.

## Figures and Tables

**Figure 1 insects-16-00543-f001:**
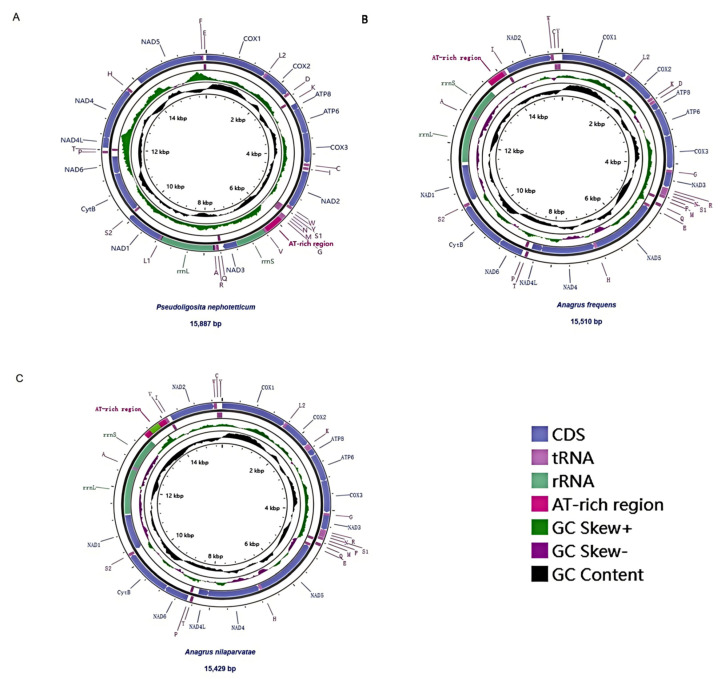
Map of the newly sequenced mitogenome. (**A**): *Pseudoligosita nephotetticum*; (**B**): *Anagrus frequens*; (**C**): *Anagrus nilaparvatae*. The genes drawn in the outermost circle are transcribed clockwise, and the genes drawn inside are transcribed counterclockwise. PCGs and rRNAs are represented by normative abbreviations, while tRNAs are indicated by single-letter abbreviations.

**Figure 2 insects-16-00543-f002:**
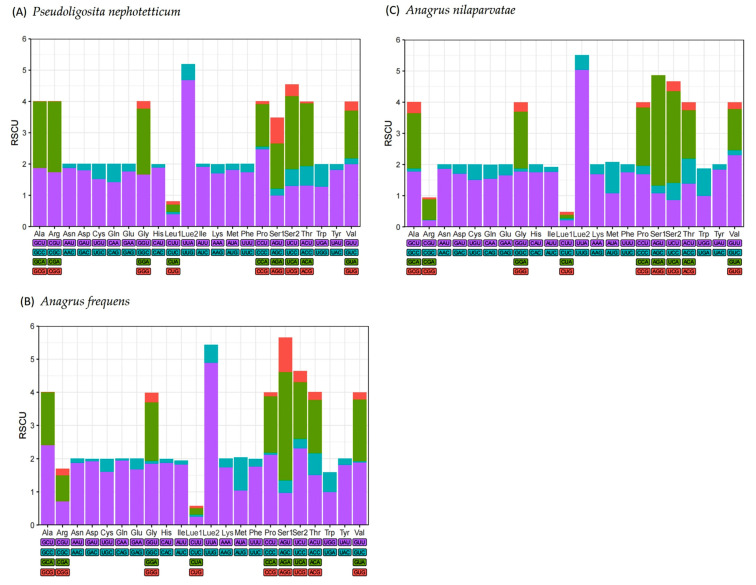
Relative synonymous codon usage of the protein-coding genes in (**A**) *P. nephotetticum*, (**B**) *A. frequens*, and (**C**) *A. nilaparvatae* mitogenome.

**Figure 3 insects-16-00543-f003:**
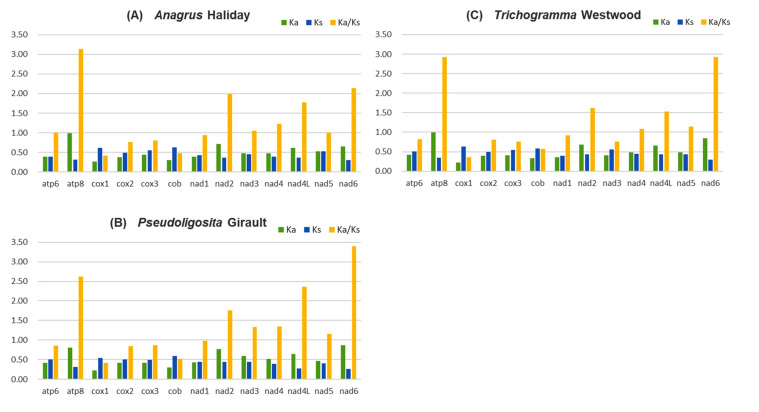
The evolution rate of 13 PCGs of 11 species from (**A**) *Anagrus*, (**B**) *Pseudoligosita*, and (**C**) *Trichogramma*. Ka refers to nonsynonymous nucleotide substitutions, Ks refers to synonymous nucleotide substitutions, and Ka/Ks refers to the selection pressure of each PCG.

**Figure 4 insects-16-00543-f004:**
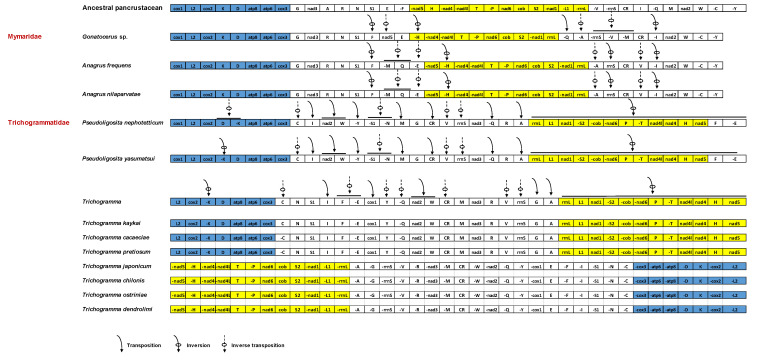
Comparison on gene arrangement of mitogenomes of *Anagrus* and *Pseudoligosita* with ancestral species. The tRNA genes are indicated by single letter amino acid codes; L1, L2, S1, and S2 denote tRNA-Leu (CUN), tRNA-Leu (UUR), tRNA-Ser (AGN), and tRNASer (UCN), respectively. Genes are transcribed from left to right, except those indicated by negative sign. Partially homoplastic gene orders are shown in the same color.

**Figure 5 insects-16-00543-f005:**
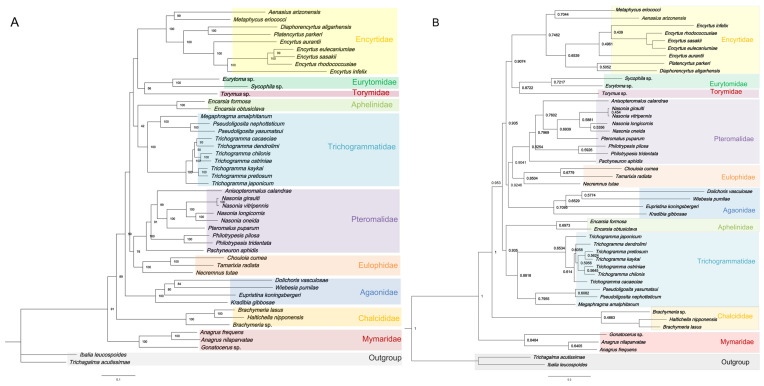
Phylogenetic trees based on 13 PCGs from 48 species. Phylogenetic trees of Chalcidoidea inferred by ML (**A**) and BI (**B**) analysis based on datasets PCG123. Numbers at the nodes are bootstrap values.

**Table 1 insects-16-00543-t001:** Nucleotide composition of the whole mitogenomes of the three egg parasitoid wasps.

Species	Feature	A %	T (U) %	C %	G %	AT %	GC %	AT Skew	GC Skew
*P. nephotetticum*	Whole genome	38.7	47.9	5.3	8.0	86.6	13.4	−0.11	0.20
	Protein-coding genes	36.3	48.6	6.4	8.7	84.9	15.1	−0.15	0.15
	1st condon site	37.7	47.6	5.6	9.2	85.2	14.8	−0.12	0.25
	2nd condon site	31.6	50.1	8.3	10.0	81.7	18.3	−0.23	0.09
	3rd condon site	39.6	48.3	5.4	6.8	87.8	12.2	−0.10	0.12
	tRNA genes	45.2	43.9	4.4	6.5	89.1	10.9	0.02	0.19
	rRNA genes	43.8	44.7	4.2	7.2	88.6	11.4	−0.01	0.26
*A. frequens*	Whole genome	42.1	44.8	6.2	7.0	86.9	13.1	−0.03	0.06
	Protein-coding genes	36.9	48.4	6.6	8.2	85.2	14.8	−0.13	0.11
	1st condon site	38.7	46.0	9.4	6.2	84.7	15.6	−0.09	0.21
	2nd condon site	34.0	48.0	9.4	8.6	82.0	18.0	−0.17	0.04
	3rd condon site	37.9	51.0	6.0	5.1	88.9	11.1	−0.15	0.08
	tRNA genes	46.1	44.9	3.4	5.6	91.0	9.0	0.01	0.24
	rRNA genes	45.3	46.0	3.7	5.0	91.3	8.7	−0.01	0.15
*A. nilaparvatae*	Whole genome	40.8	46.1	6.2	6.9	86.9	13.1	−0.06	0.05
	Protein-coding genes	36.8	47.8	6.8	8.6	84.6	15.4	−0.13	0.12
	1st condon site	39.5	46.5	5.0	9.0	86.0	14.0	−0.08	0.29
	2nd condon site	33.0	47.7	9.1	10.2	80.7	19.3	−0.18	0.06
	3rd condon site	37.9	49.3	6.2	6.6	87.2	12.8	−0.13	0.03
	tRNA genes	46.3	44.8	3.5	5.5	91.1	8.9	0.02	0.22
	rRNA genes	50.0	41.7	3.5	4.8	91.7	8.3	0.09	0.16

## Data Availability

The datasets generated and analyzed in the current study are available in the National Center for Biotechnology Information (NCBl) repository under the accession numbers PV449103 (*A. frequens*), PV449104 (*A. nilaparvatae*), PV449102 (*P. nephotetticum*).

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
