# Peer review of "Mitogenomic Characterization and Comparative Analysis of Three Egg Parasitoid Wasps Parasitizing Nilaparvata lugens (Stål)"

_insects, 2025, doi:10.3390/insects16050543_

Round 1
Reviewer 1 Report
Comments and Suggestions for Authors
This study reports the complete mitochondrial genome of three parasitoids, which are important egg parasitoid of rice planthoppers in Asian rice-growing regions. The manuscript contains a lot of important information, it is worthwhile to increase the taxonomic diversity of Chalcidoidea, in the field of mitochondrial genomics.
However, the manuscript requires additional comparative analysis with related species to be informative and valuable enough to be read. There are a few points of consideration that I think might improve it.
-Graphical abstract: After revising the manuscript, it would be more appropriate to provide a graphical abstract in the manuscript.
-lines 50-51: “Mymarids primarily parasitize Hemipteran hosts, whereas Trichogrammatids tend to target Lepidopteran species.” the claim of this sentence is too broad.
-lines 67-69: need references.
-lines 86-87: as only three species, I suggest list key features or provide the picture of key features in supplementary materials.
-line 88-91: How many wasps were used in each sample? And how many samples have you test? For Genomic DNA extracted, also need provide a brief description.
-line 237: The gene rearrangement aspect of this study is quite valuable. However, the author only compared it to Drosophila yakuba. In the past, mitogenomics researchers made similar comparisons when there were very few genomes available in GenBank. Nowadays, there are many mitogenomes available, even from closely related hymenopteran species. Generally, it is discussed that rearrangement events are associated with evolutionary characters within a phylogenetic context. Thus, the author should be compared rearrangements with more mitogenomes.
All the subgraph of each figure need be cited in the text.
All Latin names need in italic type, but sp. need not, please check all the text.
Author Response
Comments 1:-Graphical abstract: After revising the manuscript, it would be more appropriate to provide a graphical abstract in the manuscript
Response: We appreciate this valuable suggestion and have incorporated the relevant content in the revised manuscript.
Comments 2:-lines 50-51: “Mymarids primarily parasitize Hemipteran hosts, whereas Trichogrammatids tend to target Lepidopteran species.” the claim of this sentence is too broad.
Response: Thank you for your suggestion. We agree with this comment. It has been revised in the new manuscript as follow:
Line50-53: “Mymaridae primarily parasitize hosts from Hemiptera, such as Dalbulus maidis, Pere-grinus maidis, and N. lugens [5], whereas Trichogrammatidae tend to target Lepidop-teran species, including Helicoverpa armigera, Ostrinia furnacalis and Argyroploce schistaceana [6].”.
Comments 3:-lines 67-69: need references.
Response: Thank you for your reminding. We have added it to the new manuscript with some modifications
Line69-74: “As of January 2025, nine Trichogrammatinae with fully annotated mitochondrial ge-nomes have been uploaded to NCBI, seven of which are from the genus Trichogramma[17–19], with the remaining two belonging to Megaphragma[20] and Pseudoligosita[21], respectively. In contrast, the only complete mitochondrial genome of Mymaridae currently available on NCBI is that of Gonatocerus sp.[22].”
Comments 4:-lines 86-87: as only three species, I suggest list key features or provide the picture of key features in supplementary materials.
Response:Thanks to your suggestions, we've added images of these three species to the graphical abstract, and added descriptions of their characteristics in the body text.
Line102-116: ” The body length of P. nephotetticum ranges from 0.60 to 0.66 mm. The general coloration is yellowish-brown, while the compound eyes and ocelli are black. The vertex, venation of forewings and hindwings, and posterior halves of tergites on abdominal segments 1-4 exhibit a pale yellowish-brown hue. Segments posterior to the fifth abdominal segment are distinctly brown. The entire surface of the forewings appears hyaline and devoid of pigmentation[12]. The body color of A. frequens is yellow-brown. The anten-nae are geniculate, composed of a scape, pedicel, and flagellum, with the flagellum terminating in an enlarged clava. A single line of cilia is present along both the anteri-or and posterior margins of the forewing blade. The upper surface of the forewing lacks setae. The general body color of A. nilaparvatae is yellow-brown, with the scape, pedicel, and postscutellum distinctly yellow. The forewing is broadened in the distal third and bears a mid-longitudinal row of discal cilia extending from the distal portion of the stigmal vein to the wing apex. Additional scattered cilia are present between this row and both the anterior and posterior margins, resulting in the absence of any hair-less area[23].”
Comments 5:-line 88-91: How many wasps were used in each sample? And how many samples have you test? For Genomic DNA extracted, also need provide a brief description.
Response: Thanks to your suggestions, we have made the following additions to the new manuscript.
Line126-128: “In this study, each sample consisted of a single adult wasp, and all individuals within a species were offspring of the same female. A total of 15 individuals per species were used for sequencing.”
Comments 6:-line 237: The gene rearrangement aspect of this study is quite valuable. However, the author only compared it to Drosophila yakuba. In the past, mitogenomics researchers made similar comparisons when there were very few genomes available in GenBank. Nowadays, there are many mitogenomes available, even from closely related hymenopteran species. Generally, it is discussed that rearrangement events are associated with evolutionary characters within a phylogenetic context. Thus, the author should be compared rearrangements with more mitogenomes.
Response: Thank you for the helpful suggestion. We have added Gonatocerus sp. and seven additional parasitoid species from Trichogrammatidae to our analysis, with the corresponding discussion incorporated into the revised manuscript.
Line393-406: ” The patterns observed in this study—including species from the genera Anagrus, Pseudoligosita, and Trichogramma—indicate that gene order is generally conserved within genera [53]. The variation in mitochondrial gene arrangements among different genera offers potential for the design of genus-specific or species-specific primers at unique rear-rangement sites for multiplex PCR-based identification[54]. All three genera exhibited gene arrangements distinct from the ancestral insect mitochondrial genome, with Pseudoligosita and Trichogramma (When the orientation of cox1 is set as the forward strand, seven parasitoid species from the genus Trichogramma share a consistent gene arrangement pattern, hereafter referred to as the "Trichogramma " arrangement.) showing uncommon rearrangement patterns rarely reported in Chalcidoidea[55].”
Comments 7:-All the subgraph of each figure need be cited in the text.
Response: Thank you for your suggestion. Sorry for our carelessness. We have revised the manuscript accordingly.
Comments 8:-All Latin names need in italic type, but sp. need not, please check all the text.
Response: Thank you for your suggestion. Sorry for our carelessness. We have revised the manuscript accordingly.
Reviewer 2 Report
Comments and Suggestions for Authors
This study presents the first mitochondrial genomes of three egg parasitoid wasps (Pseudoligosita nephotetticum, Anagrus frequens, and Anagrus nilaparvatae) and provides valuable insights into their genomic architecture, evolutionary adaptations, and phylogenetic relationships within Chalcidoidea. The work is well-structured, methodologically sound, and addresses an important gap in parasitoid genomics. While the manuscript merits publication, minor revisions are required to enhance clarity, rigor, and adherence to formatting standards.
- Ensure consistent italicization of genus/species names (e.g., Nilaparvata lugens, Pseudoligosita).
- Italicize gene names (e.g., nad3, atp8) and use standard nomenclature for mitochondrial genes.
- Define all abbreviations at first mention (e.g., PCGs, Ka/Ks).
- Replace ambiguous phrases like “the opposite trend” with precise descriptions (e.g., “negative AT skew and positive GC skew”).
- Elaborate on how genus-specific rearrangements (e.g., Pseudoligosita inversions) could serve as taxonomic markers. For example: “The extensive rearrangements in P. nephotetticum, contrasted with conserved gene orders in Anagrus, suggest that mitochondrial architecture may resolve cryptic species complexes in Trichogrammatidae.”
- Phylogenetic analysis. Briefly discuss why ML/BI trees recovered Aphelinidae as the sister group to Trichogrammatidae in this study, contrasting with earlier reports favoring Pteromalidae. Consider adding: “The conflicting sister-group relationships may reflect differences in taxon sampling or long-branch attraction artifacts. Future phylogenomic analyses with broader taxon representation are needed to resolve this discrepancy.”
- Conclusion. Explicitly state how mitogenomic data could improve biocontrol strategies. For example: ”The genus-specific gene rearrangements and positively selected genes identified here provide molecular tools for monitoring parasitoid populations and engineering strains with enhanced host-searching efficiency.”
Author Response
Comments 1:Ensure consistent italicization of genus/species names (e.g., Nilaparvata lugens, Pseudoligosita).
Response: Thank you for pointing this out. Due to unknown formatting issues in the previous submission, the intended italicization in the Results section did not display correctly. Sorry for our carelessness. We have carefully reviewed and corrected all formatting in the revised version.
Comments 2: Italicize gene names (e.g., nad3, atp8) and use standard nomenclature for mitochondrial genes.。
Response: We appreciate your kind reminder. For reasons unknown, the italic formatting in the Results section was not properly retained in the previous version of the manuscript. Sorry for our carelessness. We have now thoroughly reviewed and corrected all relevant formatting issues in the revised submission..
Comments 3: Define all abbreviations at first mention (e.g., PCGs, Ka/Ks).
Response: We sincerely apologize for the oversight and thank you for bringing it to our attention. Thank you for your reminder.
Comments 4: Replace ambiguous phrases like “the opposite trend” with precise descriptions (e.g., “negative AT skew and positive GC skew”).
Response: Thank you for the reminder that this is indeed a much clearer presentation and has been revised in the manuscript.
Line360-364: ” Although insect mitogenomes typically exhibit positive AT skew and negative GC skew [33], our results revealed a negative AT skew and a positive GC skew our find-ings the opposite trend, suggesting a preference for thymine (T) and guanine (G) bases in these parasitoid wasps. Similar AT and GC skew bias have been reported in other Hymenopteran species [23].”
Comments 5: Elaborate on how genus-specific rearrangements (e.g., Pseudoligosita inversions) could serve as taxonomic markers. For example: “The extensive rearrangements in P. nephotetticum, contrasted with conserved gene orders in Anagrus, suggest that mitochondrial architecture may resolve cryptic species complexes in Trichogrammatidae.”
Response: Thank you for your suggestion. We didn't think carefully enough. We have added and modified it.
Line393-406: “The patterns observed in this study—including species from the genera Anagrus, Pseudoligosita, and Trichogramma—indicate that gene order is generally conserved within genera [53]. The variation in mitochondrial gene arrangements among different genera offers potential for the design of genus-specific or species-specific primers at unique rearrangement sites for multiplex PCR-based identification[54]. All three genera exhibited gene arrangements distinct from the ancestral insect mitochondrial genome, with Pseudoligosita and Trichogramma (When the orientation of cox1 is set as the forward strand, seven parasitoid species from the genus Trichogramma share a consistent gene arrangement pattern, hereafter referred to as the "Trichogramma " arrangement.) showing uncommon rearrangement patterns rarely reported in Chalcidoidea[55]. The extensive gene rearrangements observed in Pseudoligosita, in contrast to the conserved architecture of Anagrus, suggest that mitochondrial genome organization may provide a useful tool for resolving cryptic species complexes within Trichogrammatidae.”
Comments 6: Phylogenetic analysis. Briefly discuss why ML/BI trees recovered Aphelinidae as the sister group to Trichogrammatidae in this study, contrasting with earlier reports favoring Pteromalidae. Consider adding: “The conflicting sister-group relationships may reflect differences in taxon sampling or long-branch attraction artifacts. Future phylogenomic analyses with broader taxon representation are needed to resolve this discrepancy.”
Response: Thank you very much for your valuable suggestions and kind assistance. We agree that this aspect of the analysis was insufficient and have now incorporated the relevant content into the revised manuscript. We sincerely appreciate your input.
Comments 7:Conclusion. Explicitly state how mitogenomic data could improve biocontrol strategies. For example: ”The genus-specific gene rearrangements and positively selected genes identified here provide molecular tools for monitoring parasitoid populations and engineering strains with enhanced host-searching efficiency.”
Response: Thank you for your suggestions, which we've added to our latest manuscript. Thank you for your valuable advice.
Line454-462: ”This study's identification of genus-specific gene rearrangements and positively selected genes not only reveals potential molecular markers for designing parasitoid strains with enhanced host-selection efficiency, but also explores molecular tools to investigate host-parasitoid coevolutionary dynamics, thereby facilitating rapid field identification of species, continuous population tracking, and precision selection of optimized biocontrol agents against N. lugens. In addition, these findings expand the mi-tochondrial genome resources available for Chalcidoidea, refine current phylogenetic frameworks, and offer a valuable foundation for future research on parasitoid evolu-tion, taxonomy, and biological control strategies.”
Reviewer 3 Report
Comments and Suggestions for Authors
This paper describes the mitochondrial genomes of three species: Pseudoligosita nephotetticum, Anagrus frequens, and Anagrus nilaparvatae, and investigates their phylogenetic relationships. The manuscript is well-structured in terms of gene arrangement and gene structure; however, the discussion section lacks depth and fails to adequately address key findings. It is recommended to enhance the analysis by comparing phylogenetic relationships using mitochondrial genomes, nuclear genes, or alternative methodologies.
Specific revisions are as follows:
Lines 54-57: Abbreviate the genus name where appropriate.
Lines 70-71, 85-87, 117: Add necessary spaces for clarity.
Line 74: Use italics for genus names to adhere to taxonomic conventions.
Lines 78-79, 138, 141: Cite relevant references to support claims.
Line 90-91: Remove unnecessary phrases such as "(2014), according to the manual of DNeasy Blood & Tissue Kit."
Line 97: Verify units (e.g., confirm whether "10 μL" or "10 mM" is correct).
Lines 95-101: Address formatting issues (e.g., use "2×," correct "fnal" to "final," and standardize units like "g/mL").
Lines 5-107: Clarify the methodology section, particularly regarding the rationale for selecting A. nilaparvatae.
Lines 118-131: Include analyzed data to substantiate findings.
Lines 132, 163: Correct "analysis" to "analyses" for grammatical accuracy.
Lines 142, 146: Justify the use of PCGs (123 and 12) and include a nucleotide saturation analysis if applicable. Specify model selections before Bayesian Inference (BI) and Maximum Likelihood (ML) analyses.
Lines 152-153, 156, 158, 161, 177, 182-184, 187, 190, 198, 201, 212-244, 260, 281-283: Ensure species names are italicized throughout the manuscript.
Line 170: Place Table 2 on a single page for better readability.
Figure 5: Combine BI and ML results into a single figure for improved presentation.
Lines 289-293: Delete redundant sentences.
Line 307: Discuss the loss of trnL1 and compare it with other mitochondrial genomes available in NCBI.
Line 335: Replace "sp." with the appropriate species designation.
Lines 288-349: Expand the discussion section to provide a more comprehensive comparison of phylogenetic relationships using mitochondrial genomes, nuclear genes, or other methods.
This paper describes the mitochondrial genomes of three species: Pseudoligosita nephotetticum, Anagrus frequens, and Anagrus nilaparvatae, and investigates their phylogenetic relationships. The manuscript is well-structured in terms of gene arrangement and gene structure; however, the discussion section lacks depth and fails to adequately address key findings. It is recommended to enhance the analysis by comparing phylogenetic relationships using mitochondrial genomes, nuclear genes, or alternative methodologies.
Specific revisions are as follows:
Lines 54-57: Abbreviate the genus name where appropriate.
Lines 70-71, 85-87, 117: Add necessary spaces for clarity.
Line 74: Use italics for genus names to adhere to taxonomic conventions.
Lines 78-79, 138, 141: Cite relevant references to support claims.
Line 90-91: Remove unnecessary phrases such as "(2014), according to the manual of DNeasy Blood & Tissue Kit."
Line 97: Verify units (e.g., confirm whether "10 μL" or "10 mM" is correct).
Lines 95-101: Address formatting issues (e.g., use "2×," correct "fnal" to "final," and standardize units like "g/mL").
Lines 5-107: Clarify the methodology section, particularly regarding the rationale for selecting A. nilaparvatae.
Lines 118-131: Include analyzed data to substantiate findings.
Lines 132, 163: Correct "analysis" to "analyses" for grammatical accuracy.
Lines 142, 146: Justify the use of PCGs (123 and 12) and include a nucleotide saturation analysis if applicable. Specify model selections before Bayesian Inference (BI) and Maximum Likelihood (ML) analyses.
Lines 152-153, 156, 158, 161, 177, 182-184, 187, 190, 198, 201, 212-244, 260, 281-283: Ensure species names are italicized throughout the manuscript.
Line 170: Place Table 2 on a single page for better readability.
Figure 5: Combine BI and ML results into a single figure for improved presentation.
Lines 289-293: Delete redundant sentences.
Line 307: Discuss the loss of trnL1 and compare it with other mitochondrial genomes available in NCBI.
Line 335: Replace "sp." with the appropriate species designation.
Lines 288-349: Expand the discussion section to provide a more comprehensive comparison of phylogenetic relationships using mitochondrial genomes, nuclear genes, or other methods.
Author Response
Comments 1:Lines 54-57: Abbreviate the genus name where appropriate.
Response: We were really sorry for our careless mistakes. Thank you for your reminder. We have revised the manuscript accordingly.
Comments 2 :Lines 70-71, 85-87, 117: Add necessary spaces for clarity.
Response: We were really sorry for our careless mistakes. Thank you for your reminder. We have revised the manuscript accordingly.
Comments 3 :Line 74: Use italics for genus names to adhere to taxonomic conventions.
Response: Thank you for your suggestion. We have revised the manuscript accordingly.
Comments 4 :Lines 78-79, 138, 141: Cite relevant references to support claims
Response: Thank you for your reminding. We have added it to the new manuscript with some modifications. The accession numbers of all species used to build the phylogenetic tree were already listed in Table S3 from the beginning.
Line69-74: “As of January 2025, nine Trichogrammatinae with fully annotated mitochondrial ge-nomes have been uploaded to NCBI, seven of which are from the genus Trichogramma[17–19], with the remaining two belonging to Megaphragma[20] and Pseudoligosita[21], respectively. In contrast, the only complete mitochondrial genome of Mymaridae currently available on NCBI is that of Gonatocerus sp.[22].”
Line175-177: “13 PCGs from 48 mitochondrial genomes were selected for phylogenetic analysis, including 46 Chalcidoidea species and two outgroup species from Cynipoidea: Trichagalma acutissimae and Ibalia leucospoides (Table S3)[33–35].”
Comments 5 :Line 90-91: Remove unnecessary phrases such as "(2014), according to the manual of DNeasy Blood & Tissue Kit."
Response: We appreciate your attention. It has been revised in the manuscript.
Comments 6 :Line 97: Verify units (e.g., confirm whether "10 μL" or "10 mM" is correct).
Response: Thank you for your reminding. Sorry for our carelessness. It has been revised in the manuscript.
Line128-130: “The 50 μL PCR reaction mixture contained 25 μL of 2×Gflex PCR Buffer (Mg2+, dNTP plus), 2 μL F primer (10 μM), 2 μL R primer (10 μM), 5 μL of DNA and 16 μL of water.”
Comments 7 :Lines 95-101: Address formatting issues (e.g., use "2×," correct "fnal" to "final," and standardize units like "g/mL").
Response: We appreciate your attention. Sorry for our carelessness. It has been revised in the manuscript.
Comments 8 :Lines 5-107: Clarify the methodology section, particularly regarding the rationale for selecting A. nilaparvatae.
Thank you for your reminding. We have added it to the new manuscript with some modifications.
Line88-92: “After 2–3-year laboratory-based field survey of parasitoid population dynamics re-vealed P. nephotetticum, A. frequens, and A. nilaparvatae as the predominant egg parasitoids of N. lugens in rice agroecosystems. Among them, A. nilaparvatae is considered a dominant species due to its high fecundity and short developmental period[11].”
Comments 9 :Lines 118-131: Include analyzed data to substantiate findings.
Response: We appreciate your attention. It has been revised in the manuscript.
Line150-154: “For the two Anagrus species, analyses were conducted using Gonatocerus sp. (accession number: MF776883), the only member of Mymaridae with a published complete mito-chondrial genome, as the primary reference. In the case of P. nephotetticum, P. yasumatsui (accession number: NC_082212) was used as the main reference sequence, supplemented with additional species from Trichogrammatidae for comparative analysis.”
Comments 10 :Lines 132, 163: Correct "analysis" to "analyses" for grammatical accuracy.
Response: We appreciate your attention. It has been revised in the manuscript.
Comments 11 :Lines 142, 146: Justify the use of PCGs (123 and 12) and include a nucleotide saturation analysis if applicable. Specify model selections before Bayesian Inference (BI) and Maximum Likelihood (ML) analyses.
Response: We appreciate your attention. We have added those in discussion. It has been revised in the manuscript. Specific ML and BI models for constructing phylogenetic trees have been demonstrated in table S9. The result graph of heterogeneity analysis has been supplemented in figureS5.
Line303-318: “Compared to the first and second codon positions, the third position is typically more variable because of codon degeneracy, leading to a higher rate of synonymous substi-tutions and a greater likelihood of substitution saturation. To evaluate the suitability of our datasets for phylogenetic reconstruction, we performed substitution saturation tests on both PCGs and PCG12 datasets. The results showed that for the PCGs dataset, Iss = 0.360 < Iss.c = 0.855 with P < 0.05, and for PCG12, Iss = 0.340 < Iss.c = 0.851 with P < 0.05, indicating that both datasets are appropriate for phylogenetic analysis ( Iss =Index of Substitution Saturation, Iss.c =Critical Value of the Index of Substitution Saturation). The heterogeneity test results indicated that both datasets exhibited a cer-tain degree of heterogeneity, with overall differences being relatively small. However, in a few species, PCG12 showed lower heterogeneity than PCGs. This suggests that the higher A+T content at the third codon position may have contributed to the increased sequence heterogeneity (Figure S5). Phylogenetic analyses based on the concatenated PCG123 and PCG12 datasets were performed using both Bayesian inference (BI) and maximum likelihood (ML) methods (Figure 5 A-B; Supplementary Figures S4 A-B).”
Comments 12 :Lines 152-153, 156, 158, 161, 177, 182-184, 187, 190, 198, 201, 212-244, 260, 281-283: Ensure species names are italicized throughout the manuscript.
Response: Thank you for pointing this out. Due to unknown formatting issues in the previous submission, the intended italicization in the Results section did not display correctly. Sorry for our carelessness. We have carefully reviewed and corrected all formatting in the revised version.
Comments 13 :Line 170: Place Table 2 on a single page for better readability.
Response: We appreciate your attention. It has been revised in the manuscript.
Comments 14 :Figure 5: Combine BI and ML results into a single figure for improved presentation.
Response: Thank you for your suggestion. We have added and modified it.
Comments 15 :Lines 289-293: Delete redundant sentences.
Response: We appreciate your attention. It has been revised in the manuscript.
Comments 16 :Line 307: Discuss the loss of trnL1 and compare it with other mitochondrial genomes available in NCBI.
Response: Thank you for your reminder. This is indeed an issue we have considered. However, currently available records only indicate the same trnL1 loss in Gonatocerus sp. (Mymarids), which has been cited and discussed in the text.
Comments 17 :Line 335: Replace "sp." with the appropriate species designation.
Response: Thank you for your suggestion. However, the species information retrieved from NCBI and referenced articles was not accompanied by definitive taxonomic identification by the original submitters or authors. Therefore, we are currently unable to replace "sp." with specific species names.
Comments 18 :Lines 288-349: Expand the discussion section to provide a more comprehensive comparison of phylogenetic relationships using mitochondrial genomes, nuclear genes, or other methods.
Response: Thank you for your suggestion. We have added and modified it.
Line409-433: “Phylogenetic analyses based on 13 PCGs strongly supported the monophyly of Trichogrammatidae [18,19] and Mymaridae, with Mymaridae consistently positioned at the basal lineage of Chalcidoidea. However, the sister-group relationship of Trichogrammatidae remains uncertain across studies. In our ML and BI phylogenies, Aphelinidae and Trichogrammatidae were recovered as sister groups [33,35], whereas alternative datasets have placed Trichogrammatidae as the sister group to Pteromalidae [33]. The conflicting sister-group relationships may reflect differences in taxon sampling or long-branch attraction artifacts. In recent years, with the advancement and widespread application of second- generation and third-generation sequencing technologies, increasingly large datasets have been employed to explore phylogenetic relationships within Chalcidoidea. Transcriptomic data have become a particularly valuable resource for reconstructing internal relationships in this highly diverse superfamily. Early phylogenomic studies analyzed 3,239 single-copy orthologs from 37 chalcidoid taxa and 11 outgroups, but support for deep relationships along the chal-cidoid backbone remained weak, with only the grouping of Mymaridae + (Trichogrammatidae + other Chalcidoidea) being consistently recovered [13]. Subsequent studies incorporated broader taxon sampling and more extensive transcriptomic data, yielding a more resolved topology: (Mymaridae + (((Trichogrammatidae + Eulophidae) + (Encyrtidae + Aphelinidae)) + remaining Chalcidoidea))[14]. In addition, other approaches have been attempted for phylogenetic relationships within the Chal-cidoidea, such as the construction of phylogenetic relationships in the Chalcidoidea based on ultraconserved elements (UCEs) and exons[56]. Despite these advances, many relationships within Chalcidoidea remain unresolved and sometimes conflicting, highlighting the need for continued data collection and methodological refinement to achieve a widely accepted phylogenetic framework. Future phylogenomic analyses with broader taxon representation are needed to resolve this discrepancy.”
Round 2
Reviewer 3 Report
Comments and Suggestions for Authors
The manuscript has been thoroughly revised. However, we recommend reconciling the topology of the Bayesian inference (BI) tree with that of the maximum likelihood (ML) analysis to enhance methodological consistency. Specifically, aligning these phylogenetic reconstructions would improve readers' comprehension of the evolutionary relationships presented.
Comments on the Quality of English LanguageI am not native English.
